# Grafting of Anionic Decahydro-*Closo*-Decaborate Clusters on Keggin and Dawson-Type Polyoxometalates: Syntheses, Studies in Solution, DFT Calculations and Electrochemical Properties

**DOI:** 10.3390/molecules27227663

**Published:** 2022-11-08

**Authors:** Manal Diab, Ana Mateo, Joumada El Cheikh, Zeinab El Hajj, Mohamed Haouas, Alireza Ranjbari, Vincent Guérineau, David Touboul, Nathalie Leclerc, Emmanuel Cadot, Daoud Naoufal, Carles Bo, Sébastien Floquet

**Affiliations:** 1Institut Lavoisier de Versailles, CNRS, UVSQ, Université Paris-Saclay, 78035 Versailles, France; 2Laboratory of Organometallic and Coordination Chemistry, LCIO, Faculty of Sciences I, Lebanese University, Hadath 6573, Lebanon; 3Institute of Chemical Research of Catalonia (ICIQ), The Barcelona Institute of Science and Technology, 43007 Tarragona, Spain; 4Equipe de Recherche et Innovation en Electrochimie pour l’énergie (ERIEE), Institut de Chimie Moléculaire et des Matériaux d’Orsay (ICMMO), UMR CNRS 8182, Université Paris-Sud, Université Paris-Saclay, 91405 Orsay, France; 5Institut de Chimie Physique, CNRS, UMR 8000, Université Paris-Saclay, 91405 Orsay, France; 6Institut de Chimie des Substances Naturelles, CNRS UPR2301, Université Paris-Sud, Université Paris-Saclay, 91198 Gif-sur-Yvette, France

**Keywords:** polyoxometalate, hybrid, decaborate, DFT, NMR, hydrogen evolution reaction

## Abstract

Herein we report the synthesis of a new class of compounds associating Keggin and Dawson-type Polyoxometalates (POMs) with a derivative of the anionic decahydro-*closo*-decaborate cluster [B_10_H_10_]^2−^ through aminopropylsilyl ligand (APTES) acting as both a linker and a spacer between the two negatively charged species. Three new adducts were isolated and fully characterized by various NMR techniques and MALDI-TOF mass spectrometry, notably revealing the isolation of an unprecedented monofunctionalized SiW_10_ derivative stabilized through intramolecular H-H dihydrogen contacts. DFT as well as electrochemical studies allowed studying the electronic effect of grafting the decaborate cluster on the POM moiety and its consequences on the hydrogen evolution reaction (HER) properties.

## 1. Introduction

Polyoxometalates (POMs) and POM-based materials constitute a highly versatile class of compounds rich in more than several thousand inorganic compounds, which can be finely tuned at the molecular level. Because of their stunning compositions, diversified architectures and their rich electrochemical redox behaviors, they are known to display numerous properties or applications in many domains such as supramolecular chemistry [1,2,3], catalysis [4,5,6], electro-catalysis [7,8,9], and medicine, especially when POMs are functionalized with organic groups or complexes [10,11,12,13].

On their side, hydroborates represent a wide family of anionic clusters, for which many reports demonstrated their interest in different areas, especially in the biomedical domain [14,15,16,17,18]. This property thus makes the studies of borane derivatives of a great interest. In particular, the [B_10_H_10_]^2−^ cluster offers the possibility of various selective functionalizations [19,20] leading for example to *closo*-decaborate-triethoxysilane precursor, which can be coordinated to luminescent dye doped silica nanoparticles, hence facilitating the tracing of the *closo*-decaborate drug pathway in BNCT (Boron Neutron Capture Therapy) [21,22].

Driven by the synthetic challenge that constitutes the association of two anionic species with two complementary redox characters, reductive for hydroborates and oxidative for POMs, and by the biomedical applications which could be reached by associating these two families of compounds, this study aims to find the right strategy to design such POM-borate adducts and to study their chemical properties.

In a previous paper, we demonstrated that it is possible to covalently graft decaborate clusters to an Anderson-type polyoxometalate functionalized with the well-known TRIS ligand (TRIS = tris(hydroxymethyl)aminomethane), namely [Mn^III^Mo_6_O_18_(TRIS)_2_]^3−^ [23]. Nevertheless, the compound [Mn^III^Mo_6_O_18_(TRIS-B_10_)_2_]^7−^ resulting from the coupling between both components revealed to be fragile, probably because of the rigidity of the linker and the close proximity of both anionic components. This weakness is confirmed by DFT calculations indicating an athermic or slightly exothermic process for the formation of the adducts with Anderson-TRIS hybrid POMs.

In the field of hybrid POMs, the organosilyl derivatives of vacant polyoxotungstates as [PW_9_O_34_]^9−^, [SiW_10_O_36_]^8−^, [PW_11_O_39_]^7−^, [SiW_11_O_39_]^6−^, or [P_2_W_17_O_61_]^10−^ offer large diversities of compounds exhibiting a wide panel of applications [24,25]. Among them, the divacant POM Keggin [SiW_10_O_36_]^8−^ (noted hereafter SiW_10_) and the monovacant POM Dawson [P_2_W_17_O_61_]^10−^ (noted hereafter P_2_W_17_) derivatives are probably the most used because of their stability, their topology and the richness of their electrochemical properties in reduction. In particular, by reacting with aminopropyltri(ethoxy)silane (called APTES) they can provide two very useful platforms, noted respectively **SiW_10_-APTES** and **P_2_W_17_-APTES** (see Figure 1), for elaborating functional hybrid molecular architectures.

The aim of this study is to use these two different platforms to prepare new hybrid compounds associating an anionic decaborate boron cluster (denoted hereafter B_10_) with Keggin and Dawson POM derivatives. The choice of polyoxotungstate moieties rather than Mo-based POMs is based on its stability towards reduction. The employment of a long and flexible linker as APTES is essential to tackle the challenge of combining efficiently a reduced anionic boron cluster with an anionic oxidized polyoxometalate. The use of APTES linker should limit the repulsion between the two components, while its flexibility allows more easily accommodating the two entities. Finally, as shown in Figure 1, due to monovacant and divacant characters of P_2_W_17_ and SiW_10,_ respectively, it is worth noting that the relative conformations of the chains are different. For SiW_10_-APTES, the two alkyl chains are oriented nearly in parallel, whereas the monovacancy of P_2_W_17_ imposes divergent directions for the two alkyl chains. This topology is well adapted for designing triangular or square molecular species as evidenced by Izzet et al. [2,26], and in our case, we expect that these two kinds of conformation could lead to different types of adducts incorporating B_10_ clusters. In this study, we thus report the synthesis, the full characterization in solution by various NMR techniques, the electronic, the electrochemical and the electrocatalytic properties of three new hybrid POMs. In the absence of XRD structures, DFT studies provide a fine structural description of these hybrids and rationalization of their properties.

## 2. Results and Discussion

### 2.1. Syntheses

The synthesis of hybrid POMs can be achieved through different strategies. In the present study, the best synthetic procedure to get the targeted hybrid POMs has been to react first the lacunary POMs “SiW_10_” and “P_2_W_17_” with two aminopropyltri(ethoxy)silane molecules (APTES) to give the two POM-APTES precursors (see Figure 2) of formulas (TBA)_3_H[(SiW_10_O_36_)(Si(CH_2_)_3_NH_2_)_2_O]·3H_2_O (denoted hereafter **SiW_10_-APTES**) and (TBA)_5_H[P_2_W_17_O_61_(Si(CH_2_)_3_NH_2_)_2_O]·6H_2_O, denoted hereafter **P_2_W_17_-APTES**. The syntheses of these two precursors were adapted from Mayer at al. [28] by reaction of k_8_(γ-SiW_10_O_36_)·12H_2_O or K_10_α–P_2_W_17_O_61_·20H_2_O with 3-aminopropyltriethoxy silane in presence of TBABr in H_2_O/CH_3_CN medium acidified by concentrated HCl (for more details see experimental section in Appendix A). Note that for each, the proton usually written as counter-cation is in fact probably an ammonium arm R-NH_3_^+^.

The synthetic strategy to get POM-borate adducts is then to combine the amines of these POM-APTES precursors with the reactive carbonyl of the decaborate cluster [B_10_H_9_CO]^−^ (Figure 1C) to give an amide function connecting both components. Since the boron cluster can react with water for giving a carboxylic acid and since heating the synthetic mixture above 40–50 °C led to some degradation products or to some reduction in the Dawson derivative by the hydrodecaborate cluster, reactions have been conducted at room temperature and under nitrogen atmosphere. Furthermore, the coupling reaction needs the presence of a base both to help the deprotonation of the ammonium arm(s) of the POM-APTES precursors and to trap the proton produced by the coupling reaction. A moderate and a bulky organic base, diisopropylethylamine (DIPEA), was thus used to avoid the competition with APTES for the coupling reaction with [B_10_H_9_CO]^−^.

To quickly circumscribe the optimal conditions for the synthesis of the POM-borate adducts, ^29^Si, ^31^P and ^1^H NMR titrations were conducted by varying the ratios of the three reactants [B_10_H_9_CO]^−^/POM-APTES/DIPEA (all details are given in the Appendix A).

For the [B_10_H_9_CO]^−^/**SiW_10_-APTES**/DIPEA system, the ^29^Si NMR studies in solution reveal that it is possible to modulate the coupling reaction between [B_10_H_9_CO]^−^ and POM-APTES precursors by playing on the amounts of DIPEA and of [B_10_H_9_CO]^−^. For this tri-reactants system, the successive formation of two POM-borate species identified as mono- and di-adduct compounds was demonstrated thanks to their molecular symmetries (C_s_ versus C_2v_). Besides, the crucial role of DIPEA in the reaction of [B_10_H_9_CO]^−^ with POM-APTES precursors was clearly evidenced. No reaction occurs when no base is used. NMR titration studies allowed establishing that the optimal quantity of base was two equivalents for one equivalent of [B_10_H_9_CO]^−^. The Figure 2 shows for instance the proportions of SiW_10_-derivatives determined by the integration of the different peaks obtained by ^29^Si NMR in the system **SiW_10_-APTES**/[B_10_H_9_CO]^−^/DIPEA as a function of [B_10_H_9_CO]^−^/**SiW_10_-APTES** ratio at fixed DIPEA/[B_10_H_9_CO]^−^ ratio of 2.

Starting from **SiW_10_-APTES**, it evidences first the formation of a mono-adduct, which predominates for ration B_10_/**SiW_10_-APTES** = 1, before being converted into a di-adduct. The NMR titrations studies allowed establishing that using proportions **SiW_10_-APTES**/B_10_H_9_CO/DIPEA = 1/3/6 lead to the pure di-adduct denoted **SiW_10_-diB_10_**, while using 1/1/2 ratios lead to around 80% of mono-adduct mixed with some unreacted starting POM and the di-adduct. The separation of compounds has not been possible but considering the effect of the added DIPEA amounts, we succeeded to reduce the formation of the di-adduct and thus to get the mono-adduct compound denoted **SiW_10_-monoB_10_** with a good purity by decreasing the quantity of DIPEA in the proportions **SiW_10_-APTES**/B_10_H_9_CO/DIPEA = 1/1/1.5. The Figure 3 summarizes the experimental conditions used to isolate POM-borate adducts.

Similar NMR studies were also performed in solution with the Dawson derivative **P_2_W_17_-APTES** (see Appendix A). In contrast to SiW_10_ derivatives, the formation of mono- and di-adduct of the Dawson derivative are not so separated as for SiW_10_. Therefore, we failed to isolate the mono-adduct as pure product. Nevertheless, we can obtain quantitatively the di-adduct compound in the reaction mixture when ratios **P_2_W_17_-APTES**/B_10_H_9_CO/DIPEA = 1/3/6 are used.

To summarize, the multistep coupling reactions have successfully been monitored by ^29^Si and ^31^P NMR, fully described in the Appendix A, revealing that intermediate products can be followed and isolated. From these results, we established the experimental conditions allowing to selectively synthesize with good yields the mono adduct of SiW_10_ POM and the di-adducts of both POMs as mixed TBA^+^ and DIPEAH^+^ salts, namely (TBA)_3_(DIPEAH)_3_[(SiW_10_O_36_)(B_10_H_9_CONHC_3_H_6_Si)(NH_2_C_3_H_6_Si)O]·3H_2_O denoted **SiW_10_-monoB_10_**, (TBA)_6.5_(DIPEAH)_1.5_[(SiW_10_O_36_)(B_10_H_9_CONHC_3_H_6_Si)_2_O]·2H_2_O denoted **SiW_10_-diB_10_**, and (TBA)_6_(DIPEAH)_4_[(P_2_W_17_O_61_)(B_10_H_9_CONHC_3_H_6_Si)_2_O]·3H_2_O, denoted **P_2_W_17_-diB_10_** (See Experimental Section in Appendix A for more details). All adducts were isolated as powders and were characterized by FT-IR, TGA, elemental analysis, MALDI-TOF and NMR techniques. It should be noted that to our knowledge, **SiW_10_-monoB_10_** is the first example of a POM-APTES monoadduct isolated so far from the direct synthesis. All studies in the literature usually reported di-adducts with such types of hybrid POMs [29,30,31].

### 2.2. FT-IR Spectroscopy

FT-IR spectra are given in Appendix A. The FT-IR spectra of **SiW_10_-monoB_10_**, **SiW_10_-diB_10_** and **P_2_W_17_-diB_10_** evidence that the integrity of the POM part is maintained compared to the POM-APTES precursors. Furthermore, the association of the [B_10_H_9_CO]^−^ cluster is demonstrated by the disappearance of the carbonyl CO band at 2098 cm^−1^ in the B_10_H_9_CO^−^ cluster, while the broad band located at 2464–2470 cm^−1^ typical for B-H vibration bands of the decaborate moiety within the three compounds **SiW_10_-monoB_10_, SiW_10_-diB_10_** and **P_2_W_17_-diB_10_** is significantly shifted from that observed at 2517 cm^−1^ for the [B_10_H_9_CO]^−^ precursor [22,23].

### 2.3. Characterizations by MALDI-TOF Mass Spectrometry

Mass spectrometry (MS) is a very efficient technique for the characterization of polyoxometalates in solution. In our case, we did not succeed in getting mass spectra with reasonable signal-to-noise ratio and exploitable data by the usual electrospray ESI-MS technique. On the contrary, Matrix-Assisted Laser Desorption/Ionization coupled to a Time-of-Flight mass spectrometer (MALDI-TOF) revealed to be an effective technique for hybrid POMs characterization, as shown for example by Mayer and coworkers on “SiW_10_” and “P_2_W_17_“ organosilyl derivatives [28,32]. MALDI-TOF technique is applied on samples which are diluted in a matrix solution (DCTB in our case, DCTB = Trans-2-[3-(4-ter-Butylphenyl)-2-propenylidene] malonitrile) and then co-crystallized on a conductive target. Thanks to a laser irradiation, it allows producing singly charged species (cationic or anionic) and presenting the great advantage to strongly limit the number of peaks in comparison with ESI-MS spectra, where multiply charged species are generated. In the present study, the experiments were performed in both negative and positive modes (see Appendix A for the example of SiW_10_-diB_10_). According to previous works in this field, the best results were obtained in the positive mode, although the anionic character of the POM [28,32]. Indeed, as seen in the Appendix A for SiW_10_-diB_10_, the intensity reached in the negative mode appears lower, but the number of peaks is higher as there are more degradation species. Even thought our systems are polyanionic, they are more efficiently analyzed as monocationic species resulting from adducts between POMs and counter cations such as TBA^+^ and H^+^ in our case (H^+^ coming notably from DIPEAH^+^ cations or protonated amines). Furthermore, the monocationic character of the species is confirmed in all cases by the shift between peaks in the isotopic massifs.

The precursor **SiW_10_-APTES** and the compounds **SiW_10_-monoB_10_**, **SiW_10_-diB_10_** and **P_2_W_17_-diB_10_**, were thus analyzed by this technique in the positive mode. The results are gathered in Appendix A. The full spectra and a zoom on the target compounds with a spectrum simulated with IsoPro3 software are shown in Figure 4 for SiW_10_ derivatives and in Appendix A for P_2_W_17_ ones.

As shown in Figure 4 the spectrum of the precursor **SiW_10_-APTES** (Figure 4a) displays a major peak centered at m/z 4087.3 and a minor peak at *m*/*z* 4328.3. The first peak is assigned to the monocationic species {(TBA)_3_H_2_[(SiW_10_O_36_)O(SiC_3_H_6_NH_2_)_2_](CH_3_CN)_2_(H_2_O)_8_(DCTB)_2_}^+^ (calculated *m*/*z* 4087.3), while the second peak is attributed to the species {(TBA)_4_H[(SiW_10_O_36_)O(SiC_3_H_6_NH_2_)_2_](CH_3_CN)_2_(H_2_O)_8_(DCTB)_2_}^+^ (calculated *m*/*z* 4328.7). The two peaks correspond to the expected hybrid POM associated with some TBA^+^ and H^+^ cations, some solvates and two molecules of the DCTB matrix. Note that the presence of amines on the APTES part of the POM could probably favor the formation of intermolecular interactions with solvates and DCTB molecules. Such an adduct with DCTB is also observed with the precursor **P_2_W_17_-APTES** (Appendix A) but not seen with the other POMs functionalized with B_10_ clusters. The MALDI-TOF spectrum of **P_2_W_17_-APTES** indeed exhibits a major peak corresponding to the expected precursor associated with one molecule of the DCTB matrix at *m*/*z* 6058.7 (calculated *m*/*z* 6057.9 for (TBA)_6_H[(P_2_W_17_O_61_)O(SiC_3_H_6_NH_2_)_2_](DCTB)}^+^) and a minor peak at m/z 6300.0 (calculated *m*/*z* 6299.4 for (TBA)_7_[(P_2_W_17_O_61_)O(SiC_3_H_6_NH_2_)_2_](DCTB)}^+^). The attribution of the peaks is definitely confirmed thanks to the fitting of the isotopic distribution massifs. The latter are mainly due to the isotopic distribution of the 10 or 17 tungsten atoms of the POMs, which appears consistent with the experimental spectrum (see Figure 4a and Appendix A, respectively).

The spectrum of **SiW_10_-monoB_10_** depicted in Figure 4b shows only one experimental peak at *m*/*z* 3843.2 which is perfectly consistent with the calculated mass for the monocationic product {(TBA)_4_H_3_[(SiW_10_O_36_)O(SiC_3_H_6_NH_2_)(SiC_3_H_6_NHCOB_10_H_9_)](CH_3_CN)(H_2_O)_3_}^+^ (calculated *m*/*z* 3843.1). It evidences the formation of the expected adduct **SiW_10_-monoB_10_** and thus indirectly the grafting of one (B_10_H_9_CO)^−^ cluster to SiW_10_-APTES. The simulated spectrum agrees well with the experimental data, which supports this assumption although the presence of one B_10_ cluster does not modify significantly the isotopic massif.

The MALDI-TOF spectrum of **SiW_10_-diB_10_** shown in Figure 4c displays a major peak centered at *m*/*z* 4085.8, which fully agrees with the expected di-grafted compound {(TBA)_4_H_5_[(SiW_10_O_36_)O(SiC_3_H_6_NHCOB_10_H_9_)_2_](CH_3_CN)_2_(H_2_O)_6_}^+^ (*m*/*z* calculated 4084.4) and a minor peak at *m*/*z* = 4330.2 consistent with the species {(TBA)_5_H_4_[(SiW_10_O_36_)O(SiC_3_H_6_NHCOB_10_H_9_)_2_](CH_3_CN)_3_(H_2_O)_4_}^+^ (*m*/*z* calculated 4330.8). This result confirms the formation of the expected di-grafted compound.

Finally, the case of **P_2_W_17_-diB_10_**, appears more complicated, certainly due to a higher charge of the hybrid POM (10-) and a larger surface, which both favor intermolecular interactions with solvent molecules and cations. For technical reasons, the MALDI-TOF spectrum shown in Appendix A was recorded in linear mode, which does not favor the high resolution in contrast with other compounds. The spectrum displays an intense and broad experimental peak centered at *m*/*z* 6048.1, while four smaller peaks are found, respectively, at *m*/*z* 6289.6, 6431.7, 6672.5 and 6813.8. All these peaks are consistent with di-grafted species of general formula {(TBA)_x_H_y_[(P_2_W_17_O_61_)O(SiC_3_H_6_NHCOB_10_H_9_)_2_](CH_3_CN)_z_(H_2_O)_t_}^+^ (x + y = 11, z = 0–3 and t = 5–6). Regarding the main peak, the latter appears much broader than expected for only one species. Moreover, the resolution of the isotopic massif is lost. In fact, the experimental spectrum likely corresponds to a spectra superimposition of monocationic species of general formula {(TBA)_5_H_6_[(P_2_W_17_O_61_)O(SiC_3_H_6_NHCOB_10_H_9_)_2_](CH_3_CN)_x_(H_2_O)_y_}^+^ with x ranging from 1 to 5 and y from 0 to 8 (*m*/*z* in the range 6035.70 to 6076.75). Some simulated spectra are given in Appendix A.

### 2.4. NMR Studies in Solution

In the absence of crystallographic data, the three obtained hybrid systems have been thoroughly characterized by multinuclear NMR spectroscopy in order to verify their structures in solution. ^1^H, ^11^B, ^13^C, ^15^N, ^29^Si, ^31^P, and ^183^W NMR spectra were recorded in CD_3_CN at room temperature. The data are gathered in Appendix A, while selected spectra are given in Figure 5 and Figure 6 and in Appendix A.

As shown in Figure 5a and in Appendix A, ^11^B{^1^H} NMR spectrum of [B_10_H_9_CO]^−^ undergoes a significant change upon coupling with **SiW_10_-APTES** or **P_2_W_17_-APTES**. In particular, the signal at −44.4 ppm specific for the equatorial boron atom bearing the substituent CO in [B_10_H_9_CO]^−^ (B2 atom, see Figure 1c) is strongly shifted to ca. −25 ppm in the spectra of **SiW_10_-monoB_10_**, **SiW_10_-diB_10_** and **P_2_W_17_-APTES** in agreement with the grafting of the cluster on the POM.

Concomitantly, the ^1^H NMR spectrum of the mono adduct **SiW_10_-monoB_10_**, exhibits a splitting of the signals for the three methylene groups –CH_2_- of the APTES linker, denoted a, b, c (see Figure 5b), because of the lowering of the symmetry of **SiW_10_-APTES** from C_2v_ to C_s_. In addition, a new peak at 6.12 ppm assigned to an amide function is observed. For the remaining amine function, a broad signal is observed at 7.4 ppm (d), but together with two other broad signals at 5.70 and 6.33 ppm (d’ and d’’), attributed to the amine function in a frozen configuration in which the interaction with B_10_ cluster generates two inequivalent protons as depicted in Figure 7a (DFT optimized structure). These assumptions are confirmed by ^1^H-^15^N HMBC (Heteronuclear Multiple Bond Correlation) NMR spectrum (Appendix A) revealing two ^15^N signals at −251 ppm (amide) correlated to the proton signal at 6.12 ppm and at −272 ppm (free amine) correlated to the two protons peaks at 5.70 and 6.33 ppm.

Grafting a second [B_10_H_9_CO]^−^ group on the **SiW_10_-APTES** platform allows recovering the C_2v_ symmetry and thus one set of peaks was observed for the linker and especially the protons “a”, in addition to an amide peak at 5.94 ppm (Appendix A. The signals of the amine at 7.4, 5.7, and 6.3 ppm disappear in agreement with the reaction of [B_10_H_9_CO]^−^ groups with this function. Similarly, the ^1^H NMR spectrum of **P_2_W_17_-diB_10_** compared to that of **P_2_W_17_-APTES** (Appendix A) evidences the appearance of a sharp peak at 5.94 ppm assigned to an amide function, while the signal of the free amine at 7.03 ppm in the precursor P_2_W_17_-APTES disappears.

To further confirm our assignments of the signal of the amide function, ^1^H-^1^H ROESY (Rotating frame Overhause Effect SpectroscopY) and ^13^C NMR experiments were performed on **SiW_10_-diB_10_** and **P_2_W_17_-diB_10_** (Appendix A). Cross REO peaks involving the amide proton (5.94 ppm in both compounds) and some equatorial B-H protons of the B_10_ cluster at 0.4 ppm and the protons “c” of the APTES chains can be seen in Appendix A. This demonstrates the spatial proximity between these protons that interact between each other through dipolar contacts. ^13^C NMR spectra of **SiW_10_-monoB_10_**, **SiW_10_-diB_10_,** and P_2_W_17_-diB_10_ (Appendix A) notably exhibits a signal at 203 ppm assigned to a carbon atom from an amide group, which is confirmed by 2D ^1^H-^13^C HMBC NMR spectrum of **SiW_10_-monoB_10_** evidencing a correlation between this ^13^C signal at 203 ppm and the ^1^H amide signal at 6.12 ppm. Besides, in both cases of **SiW_10_-monoB_10_** and **SiW_10_-diB_10_** this ^13^C signal appears as a poorly resolved quadruplet with a coupling constant of 95 Hz consistent with a ^1^J_13C-11B_ coupling.

Therefore, ^1^H, ^1^H-^15^N HMBC, ^1^H-^1^H ROESY, ^13^C and ^1^H-^13^C HMBC NMR experiments unambiguously confirm the formation of an amide group in our three adducts by reaction of the amines of POM-APTES precursors with the carbonyl group of the cluster [B_10_H_9_CO]^−^. The modification of the ^11^B{^1^H} NMR spectra of the boron cluster after its reaction with the POM-APTES precursors further confirms such results.

^29^Si, ^31^P and ^183^W NMR probe the POM part in compounds **SiW_10_-APTES**, **SiW_10_-monoB_10_, SiW_10_-diB_10_, P_2_W_17_-APTES** and **P_2_W_17_-diB_10_** (see Figure 6 and Appendix A). The unsymmetrical environment in the mono adduct **SiW_10_-monoB_10_** is clearly confirmed through the appearance of two peaks for Si of the different linker arms at −61.9 and −63.3 ppm, while only one signal was observed at −62.3 ppm for the symmetrical di adduct **SiW_10_-diB_10_** with only a small shift from the initial SiW_10_-APTES precursor (−62.5 ppm). In addition, for all the compounds, a single peak is observed for the Si atom in the central cavity of the SiW_10_ POM moiety which is almost not affected by the grafting of the boron clusters and the resulting changes of symmetry of the adducts (Appendix A). In case of **P_2_W_17_-APTES** and **P_2_W_17_-diB_10_**, both compounds exhibit only one signal assigned to the two equivalent Si atoms of the APTES linker (Appendix A).

The ^183^W NMR spectra of precursors and adducts are given in Figure 6. For **SiW_10_-monoB_10_** the ^183^W NMR spectrum displays five peaks of intensities 2:2:2:2:2 in agreement with the expected low C_s_ symmetry, while three resonances are observed for the di adduct **SiW_10_-diB_10_** and the initial precursor **SiW_10_-APTES** of intensities (4:2:4) consistent with their C_2v_ symmetry (Figure 6a). Figure 6b shows the ^183^W NMR spectrum of P_2_W_17_-diB_10_ which differs significantly from its precursor. Both compounds exhibit nine NMR lines of integration 2:2:2:1:2:2:2:2:2 in agreement with the expected C_s_ symmetry, but their positions are slightly changed. This is due to the modification of the P_2_W_17_ moiety induced by the grafting of the two [B_10_H_9_CO]^−^ clusters. Additionally, ^31^P NMR spectra of **P_2_W_17_-APTES** and **P_2_W_17_-diB_10_** display two signals (Appendix A), wherein one of them showed a common chemical shift, while the second exhibited a small shift from −13.4 ppm in **P_2_W_17_-APTES** to −13.6 ppm in **P_2_W_17_-diB_10_.**

In conclusion, these experiments focused on the POM part fully agree in terms of molecular symmetries with the formation of the expected mono- or di-adducts with B_10_ clusters.

### 2.5. Computational Studies

The molecular geometries of **SiW_10_-APTES**, **SiW_10_-monoB_10_**, and **SiW_10_-diB_10_**, as well as those of P_2_W_17_-APTES, P_2_W_17_-monoB_10_ and P_2_W_17_-diB_10_ were fully optimized at a DFT level including implicit solvent effects (see Figure 7 and Appendix A for computational details). We considered the most relevant plausible conformers. Firstly, regarding **SiW_10_-APTES**, it exhibits two main conformers as defined by the orientation of the two amine organic arms, which we called them open and closed forms. The small difference in their relative energy, less than 1 kcal·mol^−1^ in favor of the closed form (represented in Figure 1a), forecasted that further substitution would easily overcome any initial geometric preference in the reactants. Indeed, upon B_10_ incorporation a much more complex situation arises. For **SiW_10_-monoB_10_** we characterized five conformers, two arising from the closed reactant and three species from the open reactant form. In the most stable conformer (Figure 7a), which arise from the closed form, the decaborate moiety interacts favorably with the amine hydrogens (d’ and d’’) of the unreacted arm through strong dihydrogen contacts. In the most stable open form (Figure 7b), although interaction between arms is almost neglected, the H amide atom develops other interactions. Overall, the most stable conformer given in Figure 7a is 11 kcal·mol^−1^ below the second one (Figure 7b). All five conformers lie in a narrow 20 kcal·mol^−1^ range. For the double substituted **SiW_10_-diB_10_**, since the additional repulsion arising from the negatively charged B_10_ groups, we could only characterize two forms: an open (not shown) and a closed one (two views on Figure 7c,d). The energy difference between both species was computed to be just only 5 kcal·mol^−1^. We highlight, as dashed lines in Figure 7, those hydrogen atoms of the organic arm and the B_10_ moiety that lie close in three-dimensional space.

Due to the monovacant character of P_2_W_17_, the Si-O-Si angle of the APTES moiety grafted to the POM strongly differs from that observed for the divacant SiW_10_ POM (see Figure 1). The topology of the two arms, and thus the connectivity of the two POM-APTES derivatives, strongly differs. Therefore, as seen in Figure 1, closed form is not possible for P_2_W_17_-APTES. Only one geometry could thus be considered. Then, for the mono-substituted P_2_W_17_-monoB_10,_ two conformers were characterized, one open and one folded, the folded one being more stable by only 2.2 kcal·mol^−1^. For the di-substituted Dawson derivative, only one open shaped product could be characterized (see Figure 7e,f).

Hydrogen atoms of the decaborate moieties possess a hydride character. Consequently, they can establish hydrogen-hydrogen contacts with protic solvent or with functional groups like amines or amides. In the present structures, many H-H dihydrogen contacts between the amine organic arms from APTES moiety and hydrogen atoms from decaborate clusters were observed. For instance, for **SiW_10_-monoB_10_** (Figure 7a), the hydrogen atoms d’ and d’’ from the «free» amino group are found 2.21 and 1.99 Å far from an H atom belonging to the B_10_ cluster, which are quite short distances. This fact agrees with the couplings observed in the NMR experiments.

The thermodynamics of the formation of the mono- and di-adducts of the Keggin and the Dawson species is computed exergonic in all cases as seen in Figure 8. For **SiW_10_-APTES**, both the formation of the mono- and bi-derivative were computed exergonic, 22.1 kcal·mol^−1^ for the **SiW_10_-monoB_10_**, and 13.0 kcal·mol^−1^ for the **SiW_10_-diB_10_**. The formation of **SiW_10_-monoB_10_** is clearly favored and the strong dihydrogen contacts between amino group and the grafted B_10_ cluster undoubtedly strongly stabilize such a species compared to the di-adduct **SiW_10_-diB_10_**. For P_2_W_17_-APTES, also the two substitutions are favorable, 6.8 kcal·mol^−1^ for the first, and 7.9 kcal·mol^−1^ for the second.

The computed reaction free energies for the **SiW_10_-APTES** and P_2_W_17_-APTES derivatives are fully consistent with the experimental findings. For the keggin derivatives, the strong stabilization of the mono-adduct allows isolating both mono and di-adduct thanks to the formation of strong H-H contacts. In contrast, for the Dawson derivatives, the small difference of energies between mono- and bi-adducts (1.1 eV only) does not permit isolating the mono-adduct. Besides, an excess of [B_10_H_9_CO]^−^ (3 equivalents/POM instead of 2) is needed to get the pure di-adduct compound to avoid the formation of a mixture between mono and di-grafted adducts. A similar situation was previously obtained with Anderson-type derivatives since the mono and di-adduct of B_10_ with [MnMo_6_(Tris)_2_]^3−^ are only separated by 6 eV and it was not possible to get mono-adduct [22]. This result highlights the role of the topology of the POM-APTES compounds and their faculty to stabilize species thanks to intramolecular interactions.

Finally, DFT studies provided the frontier orbitals for each compound (see Figure 9 and Appendix A. The results obtained for Keggin and Dawson derivatives exhibit the same feature. For POM-APTES the HOMO is located on one (for **SiW_10_-APTES**) or two (**P_2_W_17_-APTES**) amines of the APTES part, while the LUMO are localized on the W atoms of the POM part. By grafting the B_10_ clusters, the LUMO levels are slightly affected. LUMO remains localized on W atoms and only minor changes in energy are observed.

Conversely, the HOMO levels are drastically modified by the introduction of B_10_ clusters. Electrons of the HOMO orbitals are now mainly localized on one grafted B_10_ cluster. Interestingly, for **SiW_10_-monoB_10_** the HOMO is delocalized between one B_10_ cluster and the amine of the second arm, which strongly interacts with the B_10_ through H-H contacts. The HOMO energy level increases in all cases within the range 0.78 to 0.92 eV. The HOMO-LUMO gaps, are thus significantly reduced upon the B_10_ grafting. Indeed, for **SiW_10_-APTES**, the gap decreases from 1.67 to 0.90 eV for the first substitution, and to 0.94 eV for the second. For **P_2_W_17_-APTES**, the gap evolves from 1.56 to 0.77 eV for the first substitution, and to 0.76 eV for the second.

### 2.6. Electrochemical Properties

The electronic spectra of compounds **SiW_10_-monoB_10_**, **SiW_10_-diB_10_** and **P_2_W_17_-diB_10_** recorded in CH_3_CN containing 0.1 mol.L^−1^ TBAClO_4_ (TBAP, tetrabutylammonium perchlorate) at room temperature and 2.10^−4^ mol.L^−1^ concentration are depicted in Appendix A.

The electronic spectra of precursors **SiW_10_-APTES** and **P_2_W_17_-APTES** display absorption bands in ultraviolet region corresponding to transition between p-orbitals of the oxo ligands and d-type orbitals centered on tungsten [33,34], while the cluster [B_10_H_9_CO]^−^ exhibits weak absorption band between 300 and 200 nm notably assigned to π−π* transitions [35]. Considering that the main contribution of the spectra comes from the LMCT band involving the W atoms and that the LUMO band centered on tungsten atoms are only slightly modified upon grafting of B_10_ cluster, no drastic changes are expected in the POM-B_10_ adducts. Indeed, the electronic spectra of **SiW_10_-monoB_10_** and **P_2_W_17_-diB_10_** match well with the sum of the spectra of **SiW_10_-APTES** or **P_2_W_17_-APTES** and one or two times that of [B_10_H_9_CO]^−^, respectively. The spectrum of **SiW_10_-diB_10_** slightly differs from the sum of the component’s spectra probably due to a larger variation of LUMO energy level from **SiW_10_-APTES** to **SiW_10_-diB_10_** and additional constraints due to the vicinity of the two boron clusters.

Since no evolution of the spectra were observed within 24 h in such a medium, the compounds appear chemically stable in these experimental conditions. The cyclic voltammograms (CVs) were thus recorded for all the SiW_10_ and P_2_W_17_ derivatives and are given in Figure 10 and in Appendix A, while the anodic and cathodic potentials are gathered in Appendix A.

As depicted in Figure 10a and Appendix A, the CV of **SiW_10_-APTES** is poorly resolved and it is difficult to identify confidently all the reduction processes corresponding to the successive reduction in W^VI^ centers into W^V^, well-known to be monoelectronic in non-aqueous solvents [36]. These waves seem nevertheless reversible with processes better resolved in oxidation. Besides, an irreversible process attributed to the oxidation of amine function of the APTES linker is also observed in oxidation around +0.452 V vs. Fc^+^/Fc (see Appendix A). As for their parent precursor, CVs of **SiW_10_-monoB_10_** and **SiW_10_-diB_10_** display poorly resolved reversible electronic transfers, which appear shifted towards more negative potentials and one irreversible oxidation process around +0.452 V vs. Fc^+^/Fc assigned to the oxidation of the remaining amine function and/or of the B_10_ cluster (see Appendix A). This behavior agrees with the increase in the charge from 4- in **SiW_10_-APTES** to 6- in **SiW_10_-monoB_10_** and 8- in **SiW_10_-diB_10_** and the electron donating character of the boron cluster [37] but does not evidence a strong electronic effect of the boron cluster on the POMs electronic properties.

The CVs of **P_2_W_17_-APTES** and **P_2_W_17_-diB_10_** are given in Figure 10b and in Appendix A and appears much more resolved than those of SiW_10_ derivatives. The CV of **P_2_W_17_-APTES** displays four reversible electronic transfers with cathodic peak potentials assigned to successive mono- or bi-electronic reductions of W^VI^ centers into W(+V) [38] and two irreversible oxidation processes at E_pa_ = +0.452 and +0.759 V, assigned to the oxidation of the terminal amine groups of the APTES linkers. Conversely to the di-adduct compound **SiW_10_-diB_10_**, the CVs of the Dawson derivative **P_2_W_17_-diB_10_** give four reversible reduction processes significantly shifted towards the more positive potential compared to **P_2_W_17_-APTES** and one irreversible reduction process at E_pc_ = −2.030 V vs. Fc^+^/Fc, which was not observed in the precursor. The opposite effect was expected. This effect probably results from a combination of a charge effect, the presence of protons (in DIPEAH+ cations) and of an electronic effect of boron cluster on P_2_W_17_ moiety but at this stage it is difficult to have a clear explanation of the contribution of all these effects which can be antagonist.

Although reduction waves in the Dawson derivatives are not very well resolved, it can be observed that the di-substituted species (green line in Figure 10) is reduced at lower potentials than **P_2_W_17_-APTES**, in agreement with the fact that the LUMO and LUMO+1 raise in energy upon B_10_ attachment. Also, the successive reduction waves seem just shifted left, which would conform with the almost constant difference in the LUMO and LUMO+1 energies along the series.

### 2.7. Electrocatalytic Properties for the Reduction in Protons into Hydrogen (HER)

Many POMs are known to catalyze protons reduction into hydrogen in aqueous or in non-aqueous conditions [7,39,40]. We verified by UV-Vis spectroscopy that [B_10_H_9_CO]^−^ and its adducts with POMs are stable in CH_3_CN in the presence of excess acetic acid (20 equivalents). In these conditions, it was interesting to study the reactivity of these compounds in regard to the electro-catalytic reduction in protons into hydrogen. The experiments were performed in CH_3_CN + 0.1 M TBAP by using acetic acid as a source of protons, and as a weak acid in such a medium (pKa = 22.3) [41].

Figure 11 and Appendix A show the evolution of CVs upon stepwise addition of acetic acid up to 20 equivalents of acid/POM for all P_2_W_17_ and SiW_10_ derivatives, respectively. For all the compounds, the addition of acetic acid, gives a new irreversible reduction wave, which grows gradually with the amount of acid, expressed as γ = [acid]/[POM]. As shown in Figure 11 and Appendix A, at a given potential of −2.2 V vs Fc^+^/Fc, a linear dependence of the catalytic current versus γ is obtained, a behavior featuring the electro-catalytic reduction of protons. However, the effect of the addition of acetic acid in the solution appears stronger for P_2_W_17_ derivatives than for SiW_10_ based compounds.

To evidence the electrocatalytic process, linear voltammetry of **P_2_W_17_-diB_10_** in the presence of 20 equivalents of acetic acid was performed and compared to similar experiments performed without catalyst or on platinum electrode (Figure 11d). We notice that in the presence of the catalyst **P_2_W_17_-diB_10_**, the current density is almost doubled and there is a 250-mV overvoltage decrease compared to the solution without catalyst. Indeed, the proton reduction with respect to platinum starts at −1.400 V vs. Fc+/Fc, while it starts at −1.750 V vs. Fc^+^/Fc with catalyst and at −2.000 V vs. Fc^+^/Fc without catalyst. Finally, the formation of hydrogen is unambiguously demonstrated by gas chromatography analysis during electrolysis performed at −2.200 V vs Fc+/Fc during 4.5 h (Appendix A.

To compare the efficiency of all compounds, the catalytic efficiency (CAT) can be estimated using Equation (1):(1)$$CAT=\frac{100*\left(J_{\left(POM+20\;eq.\;CH3COOH\right)}-J_{\left(POM\;alone\right)}\right)}{J_{\left(POM\;alone\right)}}\;$$

Table 1 summarizes the *CAT* values measured for our products at −2.2 V vs Fc+/Fc. Interestingly, the two precursors **SiW_10_-APTES** and **P_2_W_17_-APTES** exhibit similar efficiency. Also, the efficiency of **SiW_10_-monoB_10_** and **SiW_10_-diB_10_** adducts are lower than that of **SiW_10_-APTES**, while it is the opposite for **P_2_W_17_-diB_10_**, which appears much more efficient than its parent precursor, in agreement with cyclic voltammetry experiments. Indeed, a less negative reduction potential of the POM part should facilitate the electro-catalytic reduction in protons.

In terms of mechanism, three key steps have to be considered: the protonation, the reduction in the catalyst and the transfer of electron towards the protons to give dihydrogen. For protonation step, since catalysis is observed in all compounds, it must occur on the most basic sites, either on the oxo groups of the POM moiety, on the free amine groups in **SiW_10_-APTES** and **P_2_W_17_-APTES** or on boron clusters for **SiW_10_-diB_10_** and **P_2_W_17_-diB_10_**. DFT calculations evidence that the most nucleophilic sites are found on the oxo ligands of the POM parts which are consequently the preferential sites for protonation (see Figure 12 and Appendix A.

For the reduction step, as seen in Appendix A, during electrolysis, the P_2_W_17_ derivatives turned to blue as expected for the reduction in such species before returning back colorless when the current is stopped indicating that the reduced POM probably transfers electrons to protons to produce hydrogen. We understand well that if this reduction occurs at higher potential, it should favor the process. **P_2_W_17_-diB_10_** is thus logically the most efficient compound.

To sum up, even if the decaborate cluster is probably not directly involved in the HER process, it plays two indirect roles: (1) the covalent grafting on POMs increases the electronic density on the POM which should facilitate the protonation step, and (2) the covalent grafting can modifies the reduction potential of the POM moieties in POM-borate adducts, which favors the reduction step of the POM species when shifted towards more positive potentials as observed in **P_2_W_17_-diB_10_**.

## 3. Conclusions

In this work, we succeeded in combining covalently the anionic [B_10_H_9_CO]^−^ cluster with anionic SiW_10_ and P_2_W_17_ derivatives using functionalized silyl derivatives (APTES) as a linker. The coupling between the two families of anionic parts appears much stronger than that with Anderson-type POMs we previously reported [23] and detailed NMR study allowed establishing the optimized conditions for the synthesis of target compounds. Hence, the selective isolation of mono- and di-adduct compounds of boron cluster with **SiW_10_-APTES**, namely [(SiW_10_O_36_)(B_10_H_9_CONHC_3_H_6_Si)(NH_2_C_3_H_6_Si)O]^6−^ and [(SiW_10_O_36_)(B_10_H_9_CONHC_3_H_6_Si)_2_O]^8−^ was successfully achieved, while only the di-adduct [(P_2_W_17_O_61_)(B_10_H_9_CONHC_3_H_6_Si)_2_O]^10−^ was isolated with **P_2_W_17_-APTES**. To the best of our knowledge, it is the first time that a mono-adduct can be isolated directly from the synthesis by functionalization of the **SiW_10_-APTES** precursor. DFT studies supported by experimental NMR data evidenced that the formation of intramolecular H-H dihydrogen contact is the driving force for the preferred formation of the mono-adduct species and such a synthetic strategy could open the route toward the formation of hybrid POMs with two different functional groups.

All these compounds were fully characterized by multi-NMR techniques including ^1^H, ^11^B, ^13^C, ^15^N, ^29^Si, ^183^W and ^31^P as well as multi-dimensional correlations such as COSY, HMBC (^1^H-^13^C and ^1^H-^15^N) and ROESY NMR allowing focusing on each part of the adducts, i.e., POM, linker and boron cluster. These characterizations demonstrated unambiguously the formation of the targeted adducts and were also consistent with FT-IR and MALDI-TOF spectrometry data. DFT studies permitted to get optimized structures for all compounds consistent with the NMR data.

The electrochemical studies allowed studying the electronic effects of the grafting of the reducing boron cluster on some oxidized POMs with probable antagonist effect between charge effect and the variation of frontiers orbitals levels upon grafting of B_10_ cluster. Finally, electro-catalytic reduction in protons into hydrogen was evidenced for these systems, the best efficiency being obtained with **P_2_W_17_-diB_10_**. The process appears mainly effective on the POM part while the boron cluster participates only indirectly to the process.

## Figures and Tables

**Figure 1 molecules-27-07663-f001:**
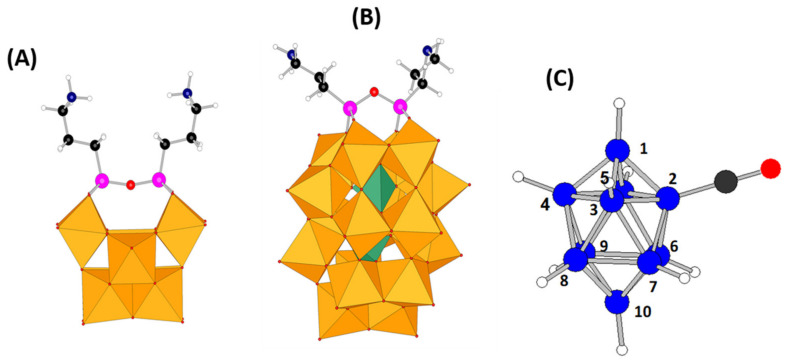
Molecular structures (DFT-optimized geometry) of (**A**) **SiW_10_-APTES** and (**B**) **P_2_W_17_-APTES** platforms highlighting the two different topologies of the APTES linker, and of (**C**) [B_10_H_9_CO]^−^ (X-ray diffraction structure from reference [27]). Legend: C in black, H in white, N in dark blue, Si in pink, O in red, B in blue, WO_6_ octahedra in orange and PO_4_ tetrahedra in green.

**Figure 2 molecules-27-07663-f002:**
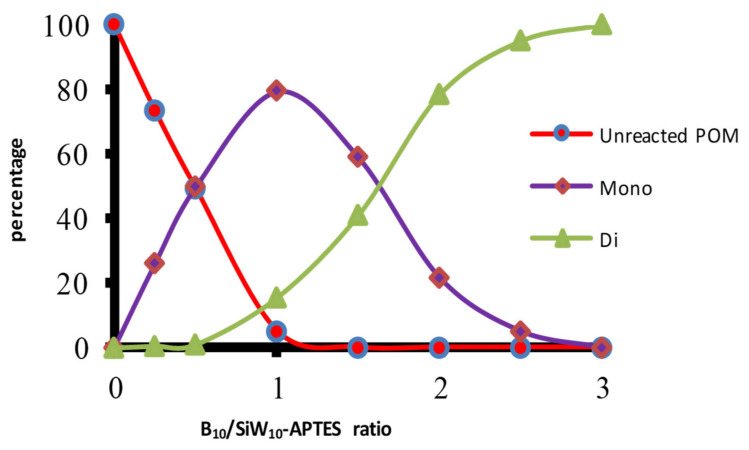
Evolution of the proportions of the products in the system **SiW_10_-APTES**/B_10_H_9_CO/DIPEA as a function of B_10_H_9_CO/**SiW_10_-APTES** ratio at fixed DIPEA/B_10_H_9_CO ratio of 2. The proportion of each species are determined by integration of the ^29^Si NMR signals. Reproduced with permission from the doctoral thesis manuscript of Dr Manal Diab, University Paris Saclay/Lebanese University, May 2018.

**Figure 3 molecules-27-07663-f003:**
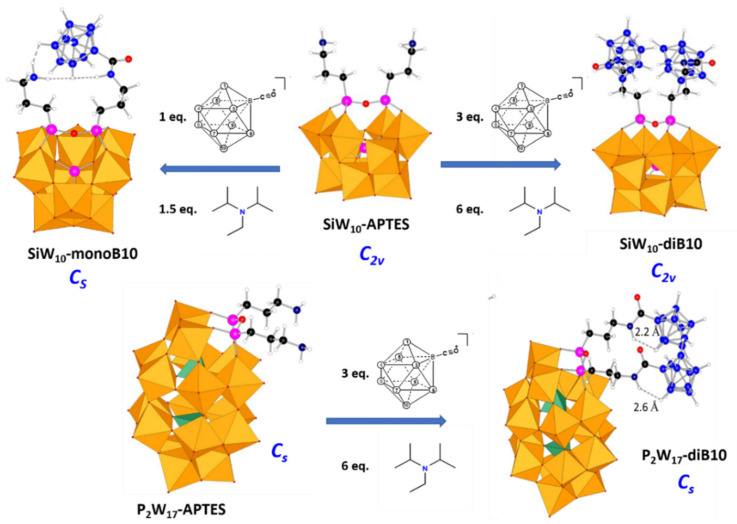
Scheme of syntheses of POM-borates adducts. The optimal quantities of reactants were determined by NMR titration studies. The reactions are performed in dry acetonitrile, at room temperature under inert atmosphere. Molecular structures are optimized geometry obtained by DFT. Legend: C in black, H in white, N in dark blue, Si in pink, O in red, B in blue, WO_6_ octahedra in orange and PO_4_ tetrahedra in green.

**Figure 4 molecules-27-07663-f004:**
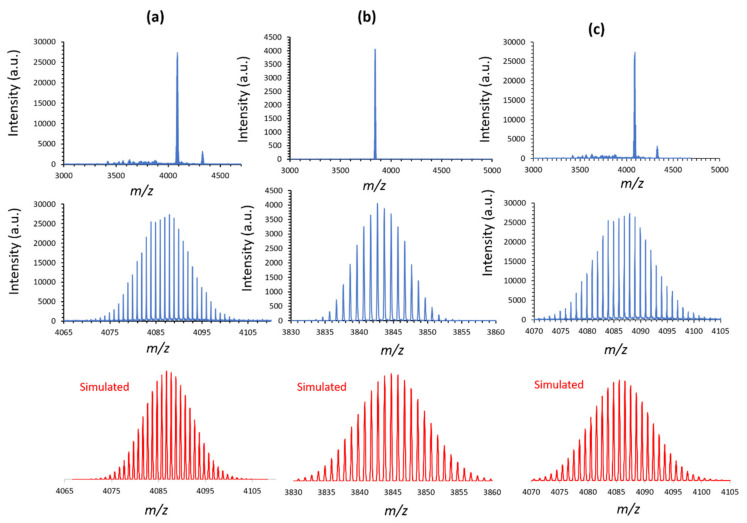
Reflector positive ion MALDI-TOF spectra of (**a**) **SiW_10_-APTES**, (**b**) **SiW_10_-monoB_10_**_,_ and (**c**) **SiW_10_-diB_10_**. Zooms of major peaks in the 3000–5000 *m*/*z* range are displayed with their respective simulated spectra.

**Figure 5 molecules-27-07663-f005:**
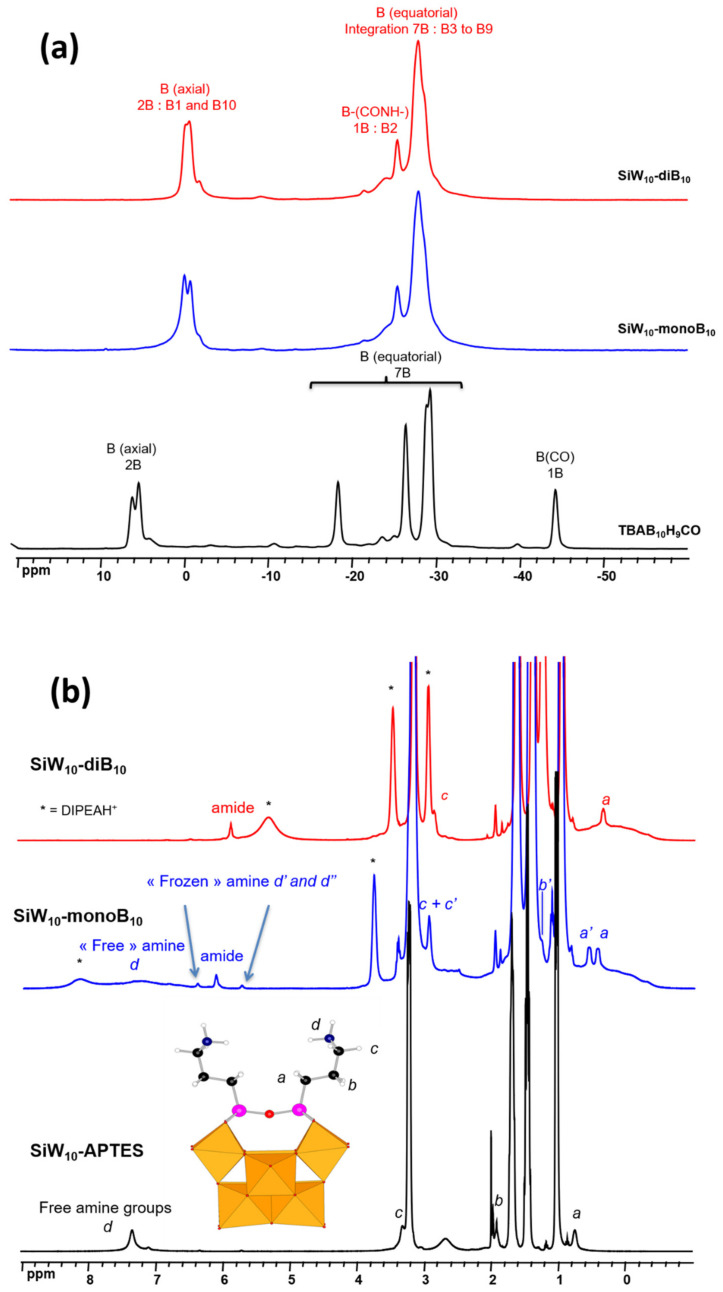
(**a**) ^11^B{^1^H} NMR spectra of **SiW_10_-monoB_10_**, **SiW_10_-diB_10_** and TBA[B_10_H_9_CO] in CD_3_CN. (**b**) ^1^H NMR spectra of **SiW_10_-monoB_10_**, **SiW_10_-diB_10_** and **SiW_10_-APTES** in CD_3_CN. * indicates the signal of the protonated amine DIPEAH+ present as a counter-cation. Reproduced with permission from the doctoral thesis manuscript of Dr Manal Diab, University Paris Saclay/Lebanese University, May 2018.

**Figure 6 molecules-27-07663-f006:**
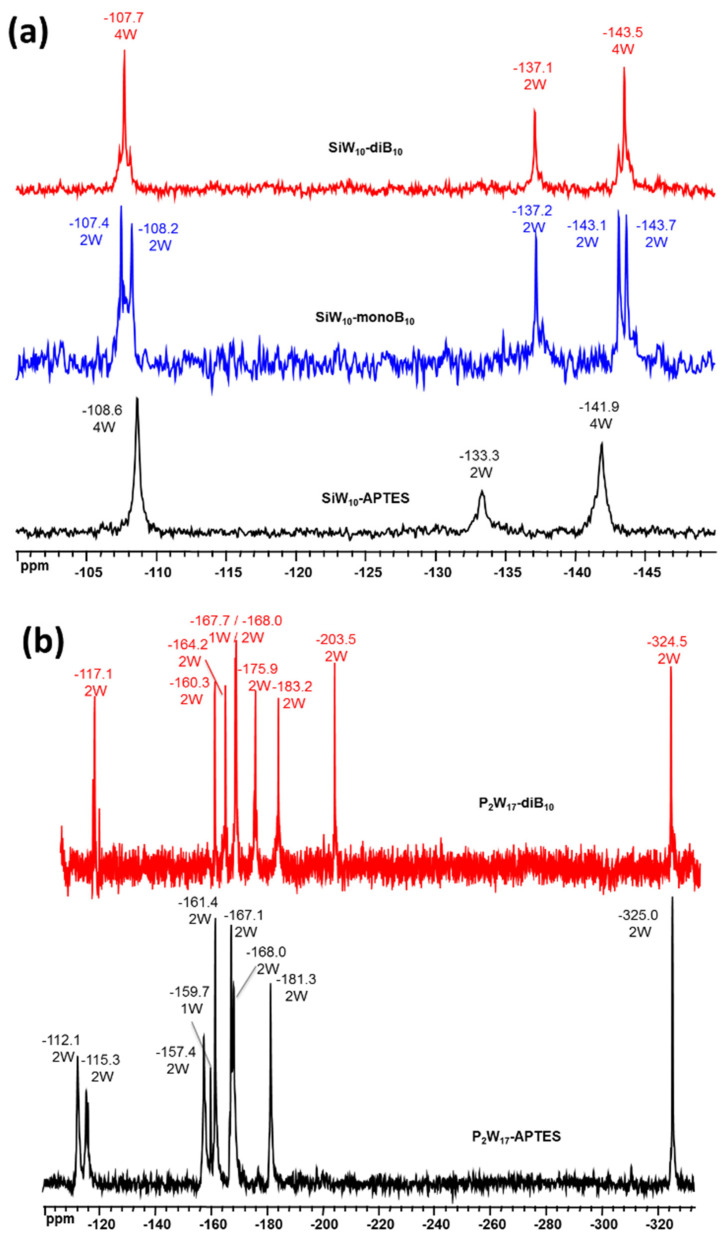
(**a**) ^183^W NMR spectra of **SiW_10_-monoB_10_**, **SiW_10_-diB_10_** and **SiW_10_-APTES** in CD_3_CN. (**b**) ^183^W NMR spectra of **P_2_W_17_-diB_10_** and **P_2_W_17_-APTES** in CD_3_CN.

**Figure 7 molecules-27-07663-f007:**
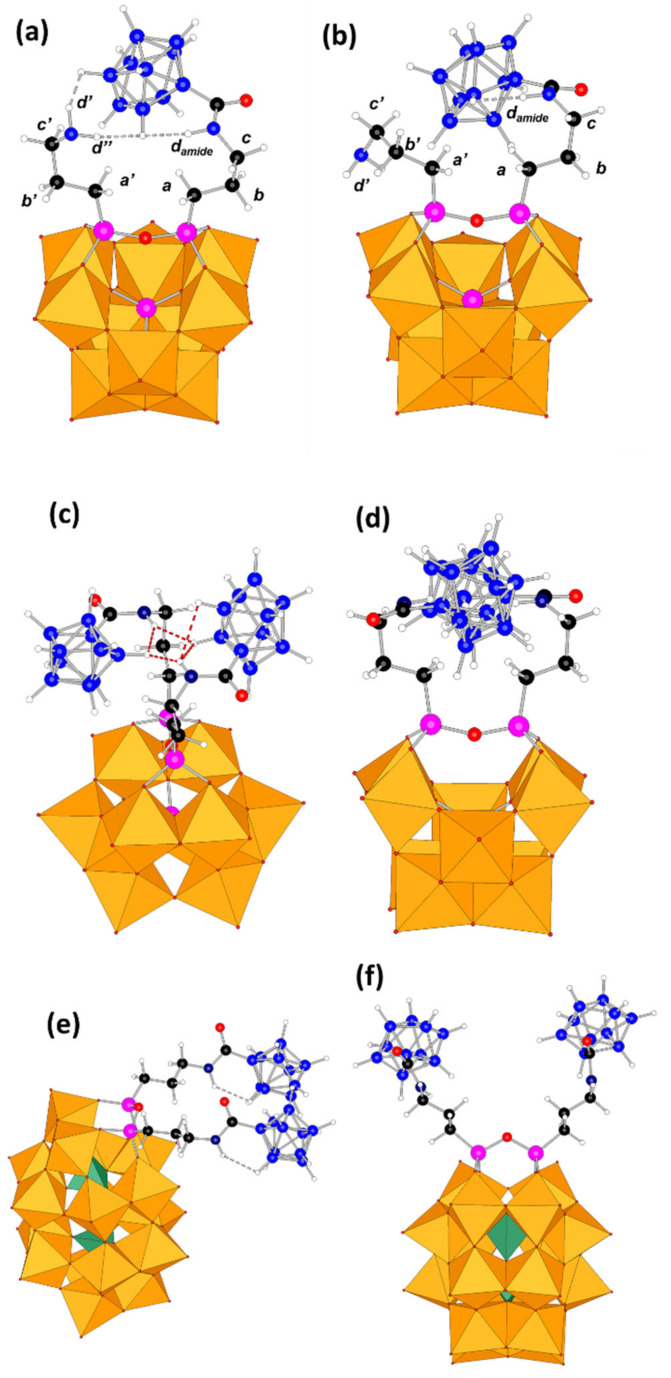
Optimized molecular structures of the POM-borates derivatives. **SiW_10_-monoB_10_** (**a**) in «closed» form and (**b**) in «open» form; (**c**,**d**) two views of the most stable configuration of **SiW_10_-diB_10_**; (**e**,**f**) two views of the most stable configuration of **P_2_W_17_-diB_10_**. Dashed lines are given for shortest H-H contacts. Legend: C in black, H in white, B in blue, N in dark blue, Si in pink, WO_6_ octahedra in orange and PO_4_ tetrahedra in green.

**Figure 8 molecules-27-07663-f008:**
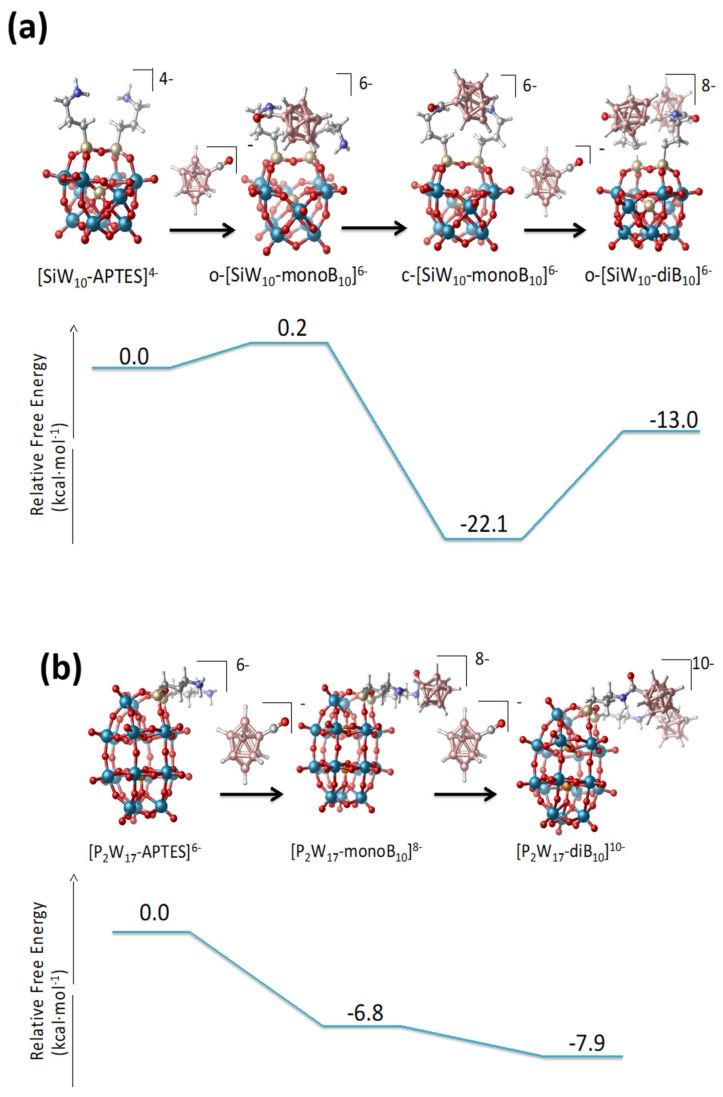
Energetic profiles of the formation of mono- and di-adduct from the starting precursors in CD_3_CN. (**a**) **SiW_10_-APTES**, **SiW_10_-monoB_10_** (open and closed isomers), and **SiW_10_-diB_10_;** (**b**) **P_2_W_17_-APTES**, **P_2_W_17_-monoB_10_**, and **P_2_W_17_-diB_10._**.

**Figure 9 molecules-27-07663-f009:**
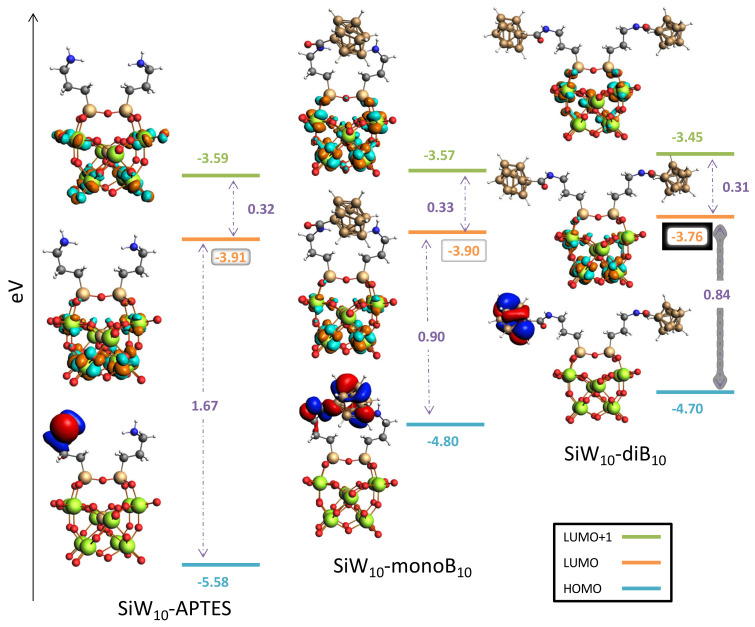
Frontier orbitals energies and energy gaps (eV) for the **SiW_10_-APTES**, **SiW_10_-monoB_10_** and **SiW_10_-diB_10_** species. Color code: W green, O red, Si light brown, B dark brown, C grey, N blue, H white; HOMO: red/blue; LUMO: orange/cyan. MO surfaces plotted at a 0.03 isovalue.

**Figure 10 molecules-27-07663-f010:**
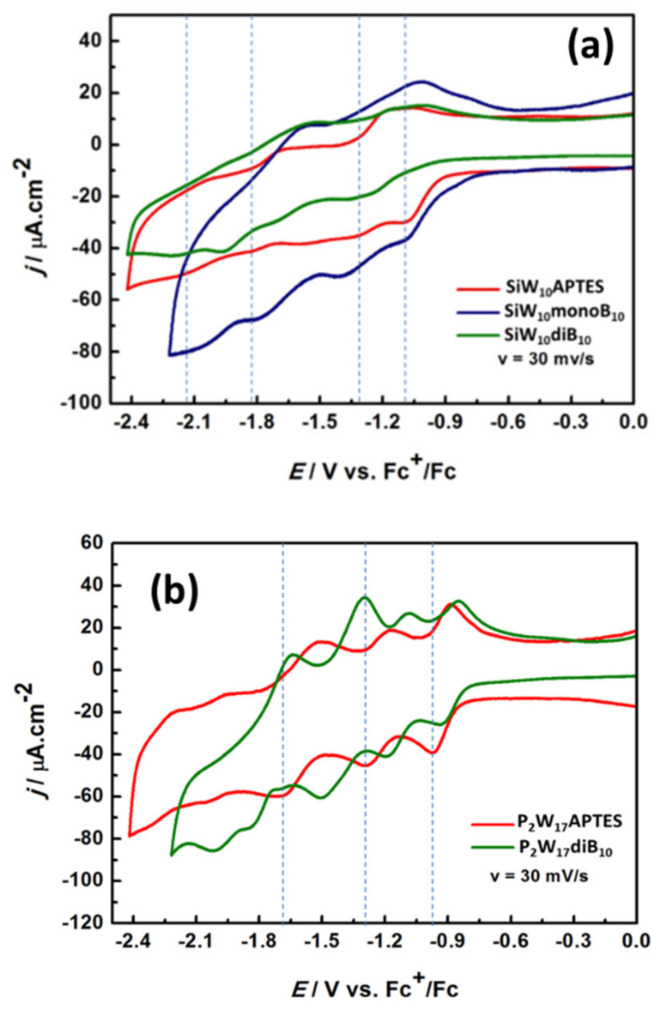
Comparison of cyclic voltammograms (**a**) for the three compounds **SiW_10_-APTES**, **SiW_10_-monoB_10_** and **SiW_10_-diB_10_**, and (**b**) for **P_2_W_17_-APTES** and **P_2_W_17_-diB_10_** in the reduction part. The electrolyte was CH_3_CN + 0.1 M TBAClO_4_. Dashed lines are only guide for eyes.

**Figure 11 molecules-27-07663-f011:**
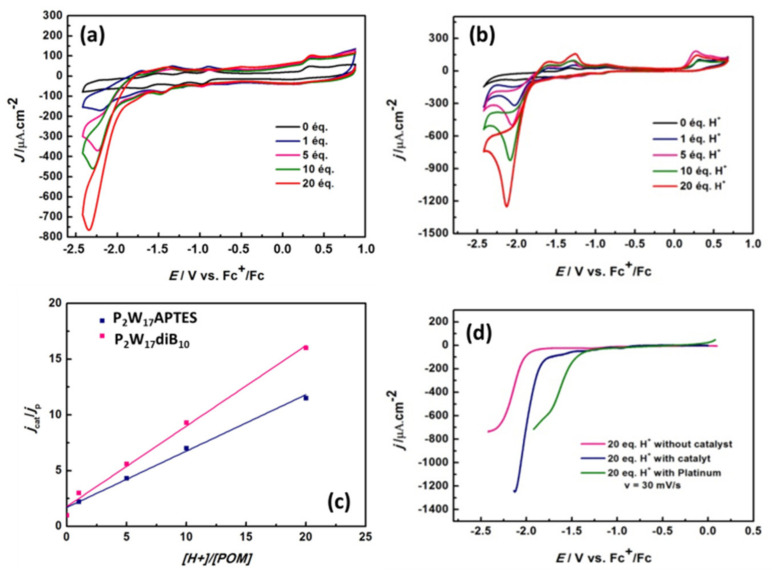
Cyclic voltammograms of (**a**) **P_2_W_17_-APTES** and (**b**) **P_2_W_17_-diB10** after addition of variable amounts of acetic acid. (**c**) Plots of the cathodic currents measured at −2.2 V vs Fc+/Fc as a function of the ratio [acid]/[POM] for **P_2_W_17_-APTES** and **P_2_W_17_-diB10**. (**d**) Comparison of HER with and without catalyst **P_2_W_17_-diB10**, and with platinum after addition of an excess of acetic acid corresponding to the quantity added for a ratio [acid]/[POM] = 20. In all cases, the electrolyte was CH_3_CN + 0.1 M TBAClO_4_. The reference electrode was a saturated calomel electrode (SCE). Reproduced with permission from the doctoral thesis manuscript of Dr Manal Diab, University Paris Saclay/Lebanese University, May 2018.

**Figure 12 molecules-27-07663-f012:**
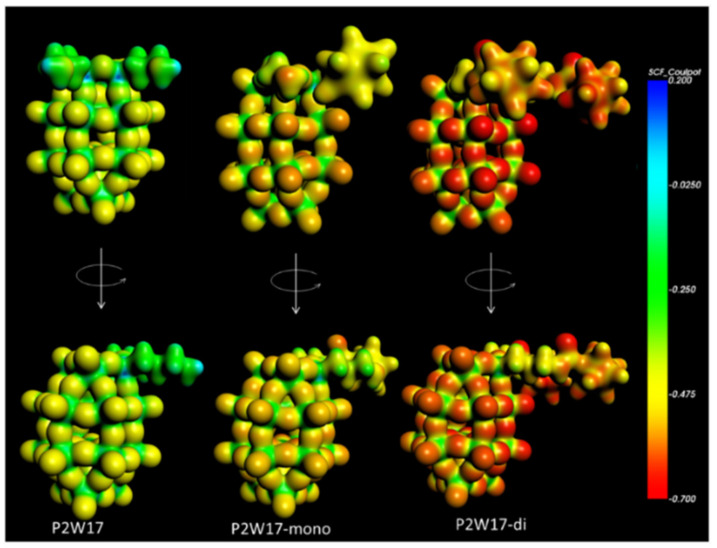
Two views of the molecular electrostatic potential in atomic units (a.u.) projected onto an electron density isosurface (0.03 e·au^−3^) for **P_2_W_17_-APTES**, **P_2_W_17_-monoB_10_** and **P_2_W_17_-diB_10_** species.

**Table 1 molecules-27-07663-t001:** Electrocatalytic efficiency for the reduction of protons into hydrogen at E = −2.2 V vs. Fc+/Fc for 20 equivalents of CH_3_COOH added in CH_3_CN.

Compound	Catalytic Efficiency (%)
**SiW_10_-APTES**	827
**SiW_10_-monoB_10_**	524
**SiW_10_-diB_10_**	548
**P_2_W_17_-APTES**	854
**P_2_W_17_-diB_10_**	1340

## Data Availability

A data set collection of computational results is available in the ioChem-BD repository and can be accessed via https://dx.doi.org/10.19061/iochem-bd-1-217 (accessed on 1 November 2022).

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
