# Peer review of "Grafting of Anionic Decahydro-Closo-Decaborate Clusters on Keggin and Dawson-Type Polyoxometalates: Syntheses, Studies in Solution, DFT Calculations and Electrochemical Properties"

_molecules, 2022, doi:10.3390/molecules27227663_

Round 1
Reviewer 1 Report
This work by Naoufal, Floquet and coworkers about covalently attaching anionic borate species to anionic polyoxometalate clusters is sound and opens new paths for the functionalization of the latter species. Isolation of a stable mono-functionalized cluster is noteworthy and all compounds reported are characterized to the greatest extent considering that no crystallographic studies could be performed. Minor suggestions/corrections are:
1) abstract: I would suggest including "H···H" before "dihydrogen contacts"
2) Introduction, page 2, lines 73-74: "The employment of a long...is essential..." instead of "...are essential..."
3) Introduction, page 2, line 80: I am not sure whether "...are more or less paralleled orientated" is correct, I would suggest "...are oriented nearly in parallel" for example
4) Syntheses, page 3, 1st paragraph: As pointed out in the supporting info, the procedure for the preparation of the POM-APTES precursors is not new, but an adaptation of a previous protocol by Mayer et al. Please, note this fact in the maintext and give the corresponding reference. A brief summary of the reaction conditions (as done for the B10 adducts) is encouraged
5) Figure 2: In my opinion, Fig 2 does not contribute to the paper substantially (no reaction conditions are given for being considered a synthetic scheme and the molecular depictions of the POM-APTES species are already inlcuded as Fig 1). I would suggest removing it. Another option could be complementing the figure with the synthesis of the B10 adducts from each POM-APTES species and including synthetic conditions
6) maldi-tof, page 6, 1st paragraph: please revise the paragraph to include a number of missing commas. In this paragraph:
6.1. line 198-199: "...characterization of polyoxometalates in solution"
6.2. line 199: "...we did not succeed in getting..." sounds better to me than "we did not succeed to get.." (also in page 10, line 336; page 18, line 596)
6.3. line 205: "in a matrix solution" sounds better to me than "in a solution of matrix"
6.3. line 219: "in all cases"
7) maldi-tof, page 6, 2nd paragraph: "The full spectra and a zoom on the ..." better than "The full spectra, a zoom on the.....". Please provide a reference for the IsoPro3 software
8) page 7:
8.1 line 235: Please correct "such an adducts with DCTB is..." either to "Such an adduct with DCTB is" or "Such adducts with DCTB are"
8.2 line 255: Please correct "The MALDI-TOF spectrum...displays a major peak..., which fully agree with..." to "The MALDI-TOF spectrum...displays a major peak..., which fully agrees with..."
8.3 line 262: I think "solvent molecules" might be better than "solvents"
9) Figure 6: It shows the word "equaterial", I guess it is a typo that should be corrected. Also, there is a type of numbering (1B, 7B, etc..) that does not match any numbering in the paper, it is somewhat confusing. It should be either removed or replaced with the numbering in Fig 1c (in this way, it would match what is written in the previous paragraph".
Moreover, I would merge this fig with fig s24: the text in the first and second paragraphs of page 9 discuss the 1H-NMR spectra and mention a,b,c-type protons, which can be somewhat hard to follow without having the spectra (and associated figure with such nomenclature) right in front
10) page 9: Please provide definitions of HMBC and ROESY abbreviations for the non-experts in NMR techniques. In line 322, "...can be seen on Figures..." should be corrected to "...can be seen in Figures..."
11) Section 2.4: The format in which the calls to the references are given is different form other sections, please homogenize. In page 12:
11.1. line 391-392: "the Si-O-Si angles....differs.." should be corrected to "the Si-O-Si angle....differs.."
11.2. line 405: "This agree..." should be corrected to "This agrees..." ("This fact agrees..." sounds better to me)
11.3 line 424: "separated by 6 eV" instead of "separated from 6 eV"?
12. Page 14, line 465: there has been a problem with the transcription of some symbols into the pdf format, lease check
Author Response
The reviewer 1 is acknowledged for his/her comments and suggestions, which were considered to prepare the revision of our paper.
This work by Naoufal, Floquet and coworkers about covalently attaching anionic borate species to anionic polyoxometalate clusters is sound and opens new paths for the functionalization of the latter species. Isolation of a stable mono-functionalized cluster is noteworthy and all compounds reported are characterized to the greatest extent considering that no crystallographic studies could be performed. Minor suggestions/corrections are:
- abstract: I would suggest including "H···H" before "dihydrogen contacts"
Reply : Good suggestion. The correction has been done
- Introduction, page 2, lines 73-74: "The employment of a..is essential..." instead of "...are essential..."
Reply : Done
- Introduction, page 2, line 80: I am not sure whether "...are more or less paralleled orientated" is correct, I would suggest "...are oriented nearly in parallel" for example
Reply : Thank you for the suggestion. The correction has been done
- Syntheses, page 3, 1st paragraph: As pointed out in the supporting info, the procedure for the preparation of the POM-APTES precursors is not new, but an adaptation of a previous protocol by Mayer et al. Please, note this fact in the maintext and give the corresponding reference. A brief summary of the reaction conditions (as done for the B10 adducts) is encouraged
Reply : We agree. A very short paragraph has been added.
- Figure 2: In my opinion, Fig 2 does not contribute to the paper substantially (no reaction conditions are given for being considered a synthetic scheme and the molecular depictions of the POM-APTES species are already inlcuded as Fig 1). I would suggest removing it. Another option could be complementing the figure with the synthesis of the B10 adducts from each POM-APTES species and including synthetic conditions
Reply : We agree. The figure 2 has been moved to the supporting information as Scheme S1.
- maldi-tof, page 6, 1st paragraph: please revise the paragraph to include a number of missing commas.
Reply : Done.
In this paragraph:
6.1. line 198-199: "...characterization of polyoxometalates in solution"
Reply : correction done.
6.2. line 199: "...we did not succeed in getting..." sounds better to me than "we did not succeed to get.." (also in page 10, line 336; page 18, line 596)
Reply : correction done.
6.3. line 205: "in a matrix solution" sounds better to me than "in a solution of matrix"
Reply : correction done.
6.3. line 219: "in all cases"
Reply : correction done.
- maldi-tof, page 6, 2nd paragraph: "The full spectra and a zoom on the ..." better than "The full spectra, a zoom on the.....". Please provide a reference for the IsoPro3 software
Reply : we agree. Isopro3 is in free access on internet. We now mention it with the link to access to the software in the experimental section. Isopro is available as freeware. (https://sites.google.com/site/isoproms/home)
8) page 7:
8.1 line 235: Please correct "such an adducts with DCTB is..." either to "Such an adduct with DCTB is" or "Such adducts with DCTB are"
Reply : correction done.
8.2 line 255: Please correct "The MALDI-TOF spectrum...displays a major peak..., which fully agree with..." to "The MALDI-TOF spectrum...displays a major peak..., which fully agrees with..."
Reply : correction done.
8.3 line 262: I think "solvent molecules" might be better than "solvents"
Reply : correction done.
- Figure 6: It shows the word "equaterial", I guess it is a typo that should be corrected. Also, there is a type of numbering (1B, 7B, etc..) that does not match any numbering in the paper, it is somewhat confusing. It should be either removed or replaced with the numbering in Fig 1c (in this way, it would match what is written in the previous paragraph".
Moreover, I would merge this fig with fig s24: the text in the first and second paragraphs of page 9 discuss the 1H-NMR spectra and mention a,b,c-type protons, which can be somewhat hard to follow without having the spectra (and associated figure with such nomenclature) right in front
Reply : we agree.
1B, 7B… correspond to the integration of the signals and not to labels of boron atoms. It can be confusing indeed. The figure 6 has been modified and clarified.
As suggested, i merged the figure 6 with figure S24 from the supporting information. It is a good suggestion for the clarity of the paper. I admit it was difficult to follow the paragraph on 1H NMR without this figure. Thanks for this suggestion.
10) page 9: Please provide definitions of HMBC and ROESY abbreviations for the non-experts in NMR techniques. In line 322, "...can be seen on Figures..." should be corrected to "...can be seen in Figures..."
Reply : We agree. Definitions of HMBC and ROESY have been added in the text. Heteronuclear Multiple Bond Correlation (HMBC) experiment correlates chemical shifts of two types of nuclei separated from each other with two or more chemical bonds
11) Section 2.4: The format in which the calls to the references are given is different form other sections, please homogenize.
Reply : correction done.
In page 12:
11.1. line 391-392: "the Si-O-Si angles....differs.." should be corrected to "the Si-O-Si angle....differs.."
Reply : correction done.
11.2. line 405: "This agree..." should be corrected to "This agrees..." ("This fact agrees..." sounds better to me)
Reply : correction done.
11.3 line 424: "separated by 6 eV" instead of "separated from 6 eV"?
Reply : correction done.
- Page 14, line 465: there has been a problem with the transcription of some symbols into the pdf format, lease check
Reply : We don’t see the problem in our version. The figures have been changed in this part. We hope i twill be better.
Reviewer 2 Report
This article systematically studied a new class of polyoxometalates (POMs). The POMs were linked to [B10H10]2- anion by organic anion APTES. The details of the synthesis and characterization are presented clearly. As for the computational study section, the density functional theory (DFT) calculations using BP86 exchange-correlation functional were performed. This functional is acceptable for most organometallic compounds. Since W is a period 6 element, the author has included the scalar-relativistic approximation by ZORA. An implicit solvent model was also applied. Therefore, the calculation methods are acceptable, and the corresponding results are reliable. There are still issues with the computational study section. The paper can be published if the author can revise the manuscript accordingly.
1. Figure 9. Please increase the font size of the y-axis label.
2. I assume that the free energy is derived from the electronic energy with some correction. The free energy correction is from the frequency calculations considering the contributions from partition functions to the entropy, enthalpy, and internal energies. It would be better to briefly indicate how the free energy is calculated from the electronic energy in the Computational Details section. Just brief description is enough. Not necessary to list all equations.
3. Figure 10 and Figure S33. The color of the LUMO isosurfaces is inconsistent with the HOMO isosurfaces. Additionally, the LUMO color is too close to the color of W atoms. Thus, the LUMO spatial distribution is a bit misleading. Please change the color scheme accordingly.
4. Please also indicate the isosurface value for Figure 10 and Figure S33 in their captions.
5. Since the HOMO is localized on the ligand side and the LUMO is localized on the POM framework, the electronic excitation exhibited the charge transfer (CT) character. What do the separated HOMO and LUMO distributions imply? The author could discuss more than just the MO energy levels.
6. Under Koopmans' theorem, the FMO level is strongly correlated with the reduction potential (if we neglect the solvation free energy). The author has done the cyclic voltammetry (CV) experiment. How does the FMO level compare to the red/ox potential? Is the trend the same for FMO and red/ox potential with respect to the B10 attachment?
7. Figure 13 and Figure S34. What is the unit for the contour plot? It would be better to indicate the unit on the scale bar or in the figure caption.
8. Is the electrostatic potential (ESP) projected on the van der Waals surface? Please indicate this detail in the Computational Details section or in the figure caption.
Author Response
The reviewer 2 is acknowledged for his/her comments and suggestions, which were considered to prepare the revision of our paper.
This article systematically studied a new class of polyoxometalates (POMs). The POMs were linked to [B10H10]2- anion by organic anion APTES. The details of the synthesis and characterization are presented clearly. As for the computational study section, the density functional theory (DFT) calculations using BP86 exchange-correlation functional were performed. This functional is acceptable for most organometallic compounds. Since W is a period 6 element, the author has included the scalar-relativistic approximation by ZORA. An implicit solvent model was also applied. Therefore, the calculation methods are acceptable, and the corresponding results are reliable. There are still issues with the computational study section. The paper can be published if the author can revise the manuscript accordingly.
- Figure 9. Please increase the font size of the y-axis label.
Reply: The Font size is increased in new Figure 8.
- I assume that the free energy is derived from the electronic energy with some correction. The free energy correction is from the frequency calculations considering the contributions from partition functions to the entropy, enthalpy, and internal energies. It would be better to briefly indicate how the free energy is calculated from the electronic energy in the Computational Details Just brief description is enough. Not necessary to list all equations.
Reply: We included a brief description on how the free energy was calculated in the Computational Details section of the Sup. Info.
- Figure 10 and Figure S33. The color of the LUMO isosurfaces is inconsistent with the HOMO isosurfaces. Additionally, the LUMO color is too close to the color of W atoms. Thus, the LUMO spatial distribution is a bit misleading. Please change the color scheme accordingly.
Reply: We agree. A new color scheme was applied in both Figures.
- Please also indicate the isosurface value for Figure 10 and Figure S33 in their captions.
Reply. Isovalues are included in our revision.
- Since the HOMO is localized on the ligand side and the LUMO is localized on the POM framework, the electronic excitation exhibited the charge transfer (CT) character. What do the separated HOMO and LUMO distributions imply? The author could discuss more than just the MO energy levels.
Reply: We thank the reviewer for raising this question. In the discusion of the «Electrochemical Properties » we refer to the localized character of the frontier orbitals and its consequences regarding the UV-Vis. We do not think necessary to include additional discussions.
- Under Koopmans' theorem, the FMO level is strongly correlated with the reduction potential (if we neglect the solvation free energy). The author has done the cyclic voltammetry (CV) experiment. How does the FMO level compare to the red/ox potential? Is the trend the same for FMO and red/ox potential with respect to the B10attachment?
Reply: We thank the reviewer for raising this question. We added the following paragraph:
“Although reduction waves in the Dawson derivatives are not very well resolved, it can be observed that the di-substituted species (green line in Figure 10) is reduced at lower potentials than P2W17-APTES, in agreement with the fact that the LUMO and LUMO+1 raise in energy upon B10 attachment. Also, the successive reduction waves seem just shifted left, which would conform with the almost constant difference in the LUMO and LUMO+1 energies along the series”.
- Figure 13 and Figure S34. What is the unit for the contour plot? It would be better to indicate the unit on the scale bar or in the figure caption. 8. Is the electrostatic potential (ESP) projected on the van der Waals surface? Please indicate this detail in the Computational Detailssection or in the figure caption.
Reply : We updated both Figure captions with the following sentence: “Two views of the molecular electrostatic potential colour maps in atomic units (a.u.) projected onto an electron density isosurface (0.03 e·a.u.-3) for….”
Round 2
Reviewer 2 Report
The author has added all the necessary information, including the computational details, isovalues, and discussions of the FMOs. I suggest the paper be published in its present form.